# One-Pot Route from Halogenated Amides to Piperidines and Pyrrolidines

**DOI:** 10.3390/molecules27154698

**Published:** 2022-07-22

**Authors:** Qiao Song, Sheng Wang, Xiangui Lei, Yan Liu, Xin Wen, Zhouyu Wang

**Affiliations:** 1Department of Chemistry, Xihua University, Chengdu 610039, China; 13980419487@163.com (S.W.); xiangui7@163.com (X.L.); 13730648246@163.com (Y.L.); m19982084625@163.com (X.W.); 2Asymmetric Synthesis and Chiral Technology Key Laboratory of Sichuan Province, Yibin 644000, China

**Keywords:** piperidine, pyrrolidine, synthesis, one-pot route, amide activation

## Abstract

Piperidine and pyrrolidine derivatives are important nitrogen heterocyclic structures with a wide range of biological activities. However, reported methods for their construction often face problems of requiring the use of expensive metal catalysts, highly toxic reaction reagents or hazardous reaction conditions. Herein, an efficient route from halogenated amides to piperidines and pyrrolidines was disclosed. In this method, amide activation, reduction of nitrile ions, and intramolecular nucleophilic substitution were integrated in a one-pot reaction. The reaction conditions were mild and no metal catalysts were used. The synthesis of a variety of N-substituted and some C-substituted piperidines and pyrrolidines became convenient, and good yields were obtained.

## 1. Introduction

Piperidine and pyrrolidine derivatives are important nitrogen heterocyclic structures with a wide range of biological activities [1,2,3,4,5]. For example, the first synthetic analgesic drug, pethidine, is a powerful analgesic still commonly used in clinical practice [6] (Figure 1, **1**). Domperidone is a dopamine receptor antagonist that is used to treat digestive disorders [7] (Figure 1, **2**). Levobupivacaine is clinically used as a long-acting local anesthetic that inhibits the release of γ-aminobutyric acid in the brain [8] (Figure 1, **3**). Bepridil is a long-acting calcium antagonist clinically used to treat angina pectoris, arrhythmia, and hypertension [9] (Figure 1, **4**). Buflomedil is a vasoactive drug with many pharmacological effects. It is widely used in the treatment of cerebrovascular and peripheral vascular diseases [10] (Figure 1, **5**).

The reported methods for the construction of piperidines and pyrrolidines are mainly divided into two categories. The first type is the reduction method, in which expensive metal catalysts, such as palladium and platinum, are used to catalyze the hydrogenation reduction of pyridines or pyrroles [11,12,13,14,15]. Moreover, the reaction usually needs to be conducted at a high pressure of tens to hundreds of atmospheres. The other type is the cyclization reaction, in which cyclic amines are constructed by nucleophilic substitution of primary amines with dihaloalkanes or diols [16,17]. However, dihaloalkanes are strong alkylating agents with high genotoxicity, and the methods using diols also face the problem of requiring a high reaction temperature of over 200 °C.

Amides are a class of relatively stable carbonyl compounds that do not readily undergo reactions in contrast to acyl halides, anhydrides, and esters. However, a significant body of research on the selective activation of amides to achieve powerful transformations under mild conditions has emerged in recent decades [18,19,20,21,22,23,24]. In 2017, Huang’s group reported an interesting method that involves the amide activation-induced dehydracoupling of halogenated secondary amides with alkenes and NaBH_4_-triggered tandem cyclization reaction; the method is used to efficiently construct 2-allyl piperidines and pyrrolidines [25]. In 2021, our group disclosed a route from aryl ethylamide to fused indolizidines and quinolizidines [26]. In this method, the following are integrated in a one-pot reaction: amide activation, Bischler–Napieralski reaction (B–N reaction) [27,28], imine reduction, and intramolecular nucleophilic substitution. We found that if the reaction is always controlled at a lower temperature, then the B–N reaction will be inhibited. Through this, the main product of piperidines or pyrrolidines was obtained rather than the fused indolizidines and quinolizidines. In the present study, a one-pot route that involved the use of halogenated amides to construct piperidine and tetrahydropyrrole derivatives was disclosed (Figure 1c).

## 2. Results and Discussion

Reaction conditions were optimized by using 5-chloro-*N*-(4-chlorolphenethyl)pentanamide (**6a**) as the starting material. Considering that pyridine class Lewis base plays an important role in the activation of amide, different kinds of base were first screened (Table 1, Entries 1–5). 2-Fluoropyridine (2-F-Py) was the most efficient (Table 1, Entry 4). When Lewis base was absent, the reaction still occurred, but the yield was reduced to a great extent (Table 1, Entry 6). Nevertheless, increasing the amount of 2-F-Py did not contribute to the improvement of yield (Table 1, Entry 7). The reaction temperature was subsequently investigated. When the reaction temperature was increased, the yield of the product decreased. (Table 1, Entries 8–9). This was due to the involving of B–N reaction at relatively higher temperatures, resulting in the production of some polycyclic byproducts. Different reductants, including KBH_4_, NaBH_3_CN, and NaBH(OAc)_3_, were then tested, but the outcomes were inferior to when NaBH_4_ was used (Table 1, Entries 10–12).

With the optimized reaction conditions defined, the scope of the reaction was investigated. N-phenethyl chloropentamides were tested first (Figure 2, **7a**–**7e**). Substrates with electron-donating or electron-withdrawing substitutions in the aromatic nucleus all reacted smoothly to generate corresponding N-phenethyl piperidines. The reaction system was then applied to N-benzyl amides (Figure 2, **7f**–**7h**). Various N-benzyl piperidines were obtained at moderate-to-good yields. Aliphatic chain and cyclic amide substrates adapted well under the established conditions and transformed into piperidine derivatives at good yields (Figure 2, **7i**–**7j**). Halogenated butyramides were subsequently tested. Similar to piperidine derivatives, tetrahydropyrrole derivatives with versatile substitutions were also successfully synthesized (Figure 2, **7k**–**7t**). We also attempted to synthesize C-substituted pyrrolidine. A 3-bromo-substituted pyrrolidine was successfully obtained in this method (Figure 2, **7u**).

We further attempted the synthesis of three-, four-, and seven-membered ring compounds. Unfortunately, none of these compounds were obtained, even if trace amounts were detected through mass spectrometry. The main products were uncycled secondary amines, which were confirmed by HRMS. Possibly, the ring tension made the reaction more difficult (Figure 2, **7v**). The synthesis of N-arylpyrrolidines was also difficult. The weaker nucleophilicity of arylamines than aliphatic amines may be the main factor that led to this result (Figure 2, **7w**). (See Appendix A)

A plausible mechanism is illustrated in Figure 3, with compound **7a** as an example. Amide substrate (Figure 3, **6a**) was firstly activated by Tf2O to obtain a nitrilium ion [29,30] (Figure 3, **8**), which was reduced by sodium borohydride to obtain the imide ion (Figure 3, **9**). The imide ion 9 was further reduced by sodium borohydride to obtain the halogenated secondary amine [31], and this was followed by intramolecular nucleophilic substitution to obtain piperidine product (Figure 3, **7a**). In our previous work [26], the nitrilium ion 8 was readily attacked by electrons on the benzene ring and underwent the B–N reaction to form the imine ion (Figure 3, **11**) under 40 °C. Imine ion 11 was then subjected to reduction of the C=N bond (Figure 3, **12**) followed by intramolecular nucleophilic substitution to obtain a fused indolizidine product (Figure 3, **13**). However, in this work, it was difficult for the electrons on the benzene ring to attack imine ion 11 at a low temperature, so the B–N reaction was inhibited. As a result, a piperidine was obtained other than a fused indolizidine.

## 3. Materials and Methods

### 3.1. General Informations

All solvents were distilled from appropriate drying agents prior to use. CH_2_Cl_2_ was distilled over calcium hydride under a nitrogen atmosphere and stored over 4Å MS. Tf_2_O was distilled over phosphorous pentoxide (P_2_O_5_) and was stored for no more than a week before redistilling. Flash column chromatography was performed using silica gel (300–400 mesh). ^1^H NMR and ^13^C NMR (400 and 101 MHz, respectively) spectra were recorded on a Bruker 400 MHz NMR spectrometer in CDCl_3_. ^1^H NMR chemical shifts were reported in ppm (δ) relative to tetramethylsilane (TMS) with the solvent resonance employed as the internal standard (CDCl_3_, 7.26 ppm). ^13^C NMR chemical shifts were reported in ppm from TMS with the solvent resonance as the internal standard (CDCl_3_, 77.16 ppm). HRMS data were recorded on a SCIEX X500R QTOF HRMS apparatus.

### 3.2. General Procedure A for the Synthesis of Piperidines and Pyrrolidines

Into a dry 25 mL round-bottom flask equipped with a magnetic stirring bar, the following were added successively: a secondary amide (compound **6**, 0.5 mmol, 1.0 equiv.), 10 mL of anhydrous CH_2_Cl_2_ and 2-F-Py (0.6 mmol, 1.2 equiv.) under an argon atmosphere. After being cooled to −78 °C, Tf_2_O (0.55 mmol, 1.1 equiv.) was added dropwise via a syringe, and the reaction was stirred for 30 min. Then, NaBH_4_ (1.0 mmol, 2 equiv.) and CH_3_OH (5 mL) were added under r.t. and stirred for additional 2 h. The reaction was quenched with a saturated aqueous solution of NaHCO_3_ (10 mL), and the mixture was extracted with dichloromethane (3 × 8 mL). The combined organic phase was dried over anhydrous Na_2_SO_4_, filtered and concentrated under reduced pressure. The residue was purified by flash chromatography on silica gel to give the corresponding compound.

## 4. Conclusions

In summary, we proposed a facile tandem protocol to construct piperidines and pyrrolidines. This method integrated amide activation, reduction of nitrile ions, and intramolecular nucleophilic substitution in a one-pot reaction. This method had mild reaction conditions and produced a variety of N-substituted and some C-substituted piperidines and pyrrolidines at good yields.

## Data Availability

The data presented in this study are available in Appendix A.

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
