# Peer review of "One-Pot Route from Halogenated Amides to Piperidines and Pyrrolidines"

_molecules, 2022, doi:10.3390/molecules27154698_

Round 1
Reviewer 1 Report
The manuscript “One-pot Route from Halogenated Amides to Piperidines and Pyrrolidines” by Qiao Song et al, reports the synthesis of piperidine using halogenated amides using Tf2O, base and reductant. An attempt is made to clarify the mechanism. In general, the communication is interesting from scientific point of view, it is well constructed and easily readable.
I have to main concerns.
1. In the manuscript itself the general procedure (e.g. scheme 2) is described with a set of reagents while in the supplementary information suddenly trimethylamine appears, the temperature is 0 oC. Please correct or explain.
In addition the HRMS (or HMRS as in the manuscript) in the SI are missing only values are provided.
2. Regarding the mechanism (scheme 3), could the authors provide some NMR of the suggested intermediates, probably would be a mixture of compounds but have you tried?
The recommendation is for minor corrections .
Author Response
Dear reviewer:
Thank you for your comments on our manuscript. Those comments are very helpful for revising and improving our paper. We have studied the comments carefully and made corrections which we hope meet with approval. The main corrections are in the manuscript and the responds to the comments are as follows(the replies are highlighted in red).
- In the manuscript itself the general procedure (e.g. scheme 2) is described with a set of reagents while in the supplementary information suddenly trimethylamine appears, the temperature is 0 oC Please correct or explain. In addition the HRMS (or HMRS as in the manuscript) in the SI are missing only values are provided.
Response:We are sorry for the misunderstanding caused by our lack of clarity in the description. The general procedure in supplementary information is for the synthesis of amide substrates, not for the synthesis of final products. We have used the description of “General procedure A for the synthesis of Piperidines and Pyrrolidines” and “General procedure B for the synthesis of amide substrates” to make the distinction. In addition, HMRS spectra have been added in the SI.
- Regarding the mechanism (scheme 3), could the authors provide some NMR of the suggested intermediates, probably would be a mixture of compounds but have you tried?
Response:The nitrilium ion intermediate (scheme 3, 8) has been well elaborated in previous references (Can. J. Chem. 2001, 79, 1694–1703.; Sci. China Chem. 2018, 61, 687–694.). In this work, we had tried to isolate intermediate 9 or 10 to provide some evidence of the mechanism in the case 7a. However, the rapid cyclization reaction made it very difficult to isolate intermediate 9 or 10. But in the case 7v, the ring tension made the cyclic reaction more difficult. The main products were uncycled secondary amines in this case, which were confirmed by HRMS. So we finally illustrated the plausible mechanism based on the literature and above results. Information about the main products of 7v has been added to the manuscript.
Reviewer 2 Report
Song and coworkers reported a synthetic method of piperidines and pyrrolidines from chloroamides. As authors mention in introduction, nitrogen-containing heterocycles are important in pharmaceutical sciences. Although the reported method will be useful, authors previously reported a tandem reaction using this cyclization. In this report, they exhibit monocyclized products can be obtained by controlling the reaction temperature. This reviewer think that this worked should be published as a full paper in more specific journal.
Functional group compatibilities, for example ester, ketone, nitrile and alcohols, should be investigated before publication.
In terms of mechanism, does author conform that amine 10 spontaneously cyclize to 7a. Is there any possibility the cyclization of boron complex.
Please check Figure 1, compound 3, which is not Levobupivacaine.
Author Response
Dear reviewer:
Thank you for your comments on our manuscript. Those comments are very helpful for revising and improving our paper. We have studied the comments carefully and made corrections which we hope meet with approval. The main corrections are in the manuscript and the responds to the comments are as follows(the replies are highlighted in red).
- Functional group compatibilities, for example ester, ketone, nitrile and alcohols, should be investigated before publication.
Response:We appreciate the valuable advice for the scope investigation of this method. Substrates containing ketone or nitrile were not designed in this paper because ketone was readily reduced by NaBH4 and nitrile was easy to react with the nitrilium ion intermediate (scheme 3, 8). We understand that synthesizing new compound containing more functional group such as ester and alcohol may better reveal the scope of this method. However, since it is now the summer vacation period, replenishing new compounds becomes difficult for us at this moment. Considering that we have already made derivatives of benzyl, phenylethyl, chain and cycloalkyl substitutions on N atoms, as well as different electron-withdrawing and electron-donating groups and different halogen substitution products on aromatic rings, is it possible for us not to add new compounds in this paper? Of course, we will seriously consider the reviewer's suggestions and continue to study functional group compatibilities of this method. Related content will be reflected in our later work.
- In terms of mechanism, does author conform that amine 10 spontaneously cyclize to 7a. Is there any possibility the cyclization of boron complex.
Response:Thanks very much for offering us new thoughts about the mechanism. The nitrilium ion intermediate (scheme 3, 8) was readily reduced to secondary amines by sodium borohydride (Synlett 2010, 2010, 1829–1832). Although we did not directly obtain intermediate 9 or 10 in case 7a, we found that the main products in case 7v were the uncyclized amine due to ring tension, which was confirmed by HRMS. Therefore, we still believe that the cyclization reaction is more likely to undergo amine 10. Information about the main products of 7v has been added to the manuscript.
- Please check Figure 1, compound 3, which is not Levobupivacaine.
Response:We are sorry for our mistake. Figure 1 had been corrected.
Once again, thank you very much for your constructive comments and suggestions which would help us to improve the quality of the paper.
Very sincerely yours,
Zhouyu Wang
Institution and address: Department of Chemistry, Xihua University, Chengdu, 610039, China.
Email: zhouyuwang77@163.com
Reviewer 3 Report
The manuscript “One-pot Route from Halogenated Amides to Pyperidines and 2-Pyrrolidines.” describes the synthesis of five and six membered heterocycles involving amide activation, reduction of nitrilium ion, and intramolecular nucleophilic substitution in a one-pot. This work is a different results of author’s previous work in which the aromatic ring was involved in Bischler–Napieralski cyclisation before reduction at 40 °C, whereas in present report reduction takes place initially to avoid the B-N cyclisation at room temperature. Results are interesting, even electron rich aromatic ring also not involved in the cyclisation (examples 7e and 7n). Variety of substrates are studied and mechanistic path clearly explains the chemoselectivity. In my opinion this work could be published in “Molecules” journal after addressing the following points.
1) In figure-1 compound 3 represents loperamide whereas in the explanation (Line 23) it has mentioned as Levobupivacaine, must be corrected.
2) At 0 °C product yield was decreased, is this because of the involvement of Bischler–Napieralski reaction or unreacted starting material? An explanation is required.
3) In case of products 7v, is amide reduced to secondary amine? Results may be included.
4) For compound 7c H1NMR coupling constants for aromatic region should be corrected.
Author Response
Dear reviewer:
Thank you for your comments on our manuscript. Those comments are very helpful for revising and improving our paper. We have studied the comments carefully and made corrections which we hope meet with approval. The main corrections are in the manuscript and the responds to the comments are as follows(the replies are highlighted in red).
- In figure-1 compound 3 represents loperamide whereas in the explanation (Line 23) it has mentioned as Levobupivacaine, must be corrected.
Response:We are sorry for our mistake. The structure of Levobupivacaine in Figure 1 had been corrected.
- At 0 °C product yield was decreased, is this because of the involvement of Bischler–Napieralski reaction or unreacted starting material? An explanation is required.
Response:When reaction temperature raised, the Bischler–Napieralski reaction would involve in this reaction, resulting in producing some polycyclic byproducts. The explanation has been added into the manuscript.
- In case of products 7v, is amide reduced to secondary amine? Results may be included.
Response: In case of products 7v, the main products were indeed secondary amine, which were confirmed by HRMS. The results have been included in the manuscript.
- For compound 7c H1NMR coupling constants for aromatic region should be corrected.
Response:Thanks for pointing out our mistake. coupling constants of compound 7c has been corrected.
Once again, thank you very much for your constructive comments and suggestions which would help us to improve the quality of the paper.
Very sincerely yours,
Zhouyu Wang
Institution and address: Department of Chemistry, Xihua University, Chengdu, 610039, China.
Round 2
Reviewer 2 Report
Thank you for sending me a revised manuscript. In terms of the mechanism, I can understand that the isolation of compound 10 is difficult, if the cyclization spontaneously occurs. I can understand authors' situation. However, after summer vacation, additional substrates to uncover the functional compatibility, should be investigated.